



# Synergy of millimeter-wave radar and radiometer measurements for retrieving frozen hydrometeors in deep convective systems

Keiichi Ohara[1,2], Hirohiko Masunaga[3]

[1] Earth Observation Research Center, Japan Aerospace Exploration Agency, Ibaraki, Japan
[2] Graduate School of Environmental Studies, Nagoya University, Nagoya, Japan
[3] Institute for Space-Earth Environmental Research, Nagoya University, Nagoya, Japan

*Correspondence to*: Keiichi Ohara(ohara.keiichi@jaxa.jp)

**Abstract.**

Satellite remote sensing of frozen hydrometeors in deep convective systems is essential for understanding precipitation
systems and the formation of upper-level clouds. To reduce uncertainties in ice cloud microphysical properties inside convective clouds, a combined use of millimeter-wave sensors sensitive to frozen particles in deep convective clouds is a promising strategy. This study uses the CloudSat Cloud Profiling Radar (CPR) and the Global Precipitation Measurement (GPM) Microwave Imager (GMI) to retrieve the vertical profiles of ice water content (IWC), number concentration ($N_t$) and mass-weighted diameter ($D_m$). A new retrieval method is developed by a combination of Deep Neural Network (DNN) and
Optimal Estimation Method (OEM). In the first step of the algorithm, an initial guess is estimated by DNN based on an *a priori* database, followed by the next step where OEM seeks a more optimal frozen hydrometer profile.

The retrieval performance is evaluated against selected match-up observations of CloudSat and GPM. The combined use of CPR and GMI observations reduce retrieval errors compared to the CPR-only observations. The retrieved frozen hydrometer profiles excellently reproduce CPR reflectivity and GMI brightness temperatures ($T_b$) when computed by forward
simulations. The dual-frequency precipitation radar (DPR) reflectivity is also reasonably reproduced, indicating some ability to retrieve large snow and graupel particles detectable by the low-frequency radars. Among different ice habit models tested, the optimal models for this synergistic algorithm are dendrite snowflake and soft sphere for the ice density model used in this algorithm. The combined algorithm developed by this work implies the potential of passive and active millimeter-wave instruments for retrieving multiple aspects of the cloud ice properties when combined in tandem. Future work will
incorporate new satellite missions, including EarthCARE Doppler millimeter-wave radar and submillimeter-wave radiometers such as Ice Cloud Imager.



## 1 Introduction

Frozen hydrometeors such as cloud ice, snow, and graupel play a crucial role in tropical convective cloud systems, particularly those accompanied by intense rainfall and widespread anvils. Deep convective clouds as often observed in the tropics contain a significant amount of solid precipitation particles aloft, which serve as the primary source of heavy precipitation at the surface. Moreover, cirrus anvils detrained from deep convection contributes to the formation of nearly one half of tropical upper-level clouds (Luo and Rossow, 2004). These ice clouds in the tropical upper troposphere impose a

significant radiative forcing in both shortwave and longwave spectra, and the imbalance between the shortwave and longwave effects depends on cloud microphysical properties (Hartmann and Berry, 2017; Ohno and Satoh, 2018). The radiative forcing of clouds associated with global warming is identified as one of the largest sources of uncertainty in climate change predictions (IPCC 2021). To understand the formation processes of precipitation systems in deep convective clouds and tropical upper-level clouds, it is crucial to observationally clarify the properties of frozen hydrometeors formed within

convective clouds.

Understanding the properties of frozen hydrometeors is a significant challenge for both numerical modeling and satellite observations. Among the General Circulation Models (GCMs) used in IPCC assessments, significant discrepancies in the global mean ice water path (IWP) have been reported, resulting mainly from limitations in cloud parameterization (Waliser et al., 2009). These discrepancies give rise to errors in climate predictions and uncertainties in the cloud feedbacks associated

with global warming. High-quality global-scale satellite observation data are instrumental for validating the climate models. However, the IWP estimates from satellite observations, while relatively consistent in spatial distribution, have significant discrepancies in absolute values among one another (Duncan and Eriksson, 2018; Eliasson et al., 2011). The primary sources of these discrepancies are believed to be the uncertainties in the cloud microphysical properties and differences in the sensor's sensitivity to ice particles (Duncan and Eriksson, 2018).

To reduce the uncertainty in cloud microphysical properties, a combined use of multiple sensors offers a promising strategy. The signals observed by satellite sensors depend not only on ice water content (IWC) but also on the cloud microphysical properties such as particle size distribution (PSD) and particle shape. Constraining IWC and the cloud microphysical properties at the same time benefits from a synergy of multiple sensors with different measuring principles, which could complement the technical limitations of individual sensors alone. Cloud ice observations have historically begun with

passive sensors in the visible, infrared (Heidinger and Pavolonis, 2009), and microwave spectrum (Deeter and Franklin Evans, 2000; Evans et al., 2012). In recent years, methods for a combined use of radar and lidar have been developed (Delanoë, J., and R. J. Hogan, 2008, 2010; Deng et al., 2015; Deng, M., G. G. Mace, Z. Wang, and H. Okamoto, 2010; Okamoto, 2003; Okamoto et al., 2010). Radar and lidar observations of cloud ice, independently or in tandem, has led to significant advancements in reducing the uncertainty of cloud microphysical properties. However, the synergy of radar and

lidar observations is not optimal for the retrieval of frozen hydrometer within thick clouds such as convective clouds,



because the lidar signals experience a sever attenuation. The uncertainty in cloud microphysical properties within the convective clouds remains a significant challenge.

This study explores a combined use of millimeter-wave radar and radiometer measurements, which are both able to penetrate through a deep cloud layer better than lidar observations. The Cloud Profiling Radar (CPR) aboard the CloudSat satellite and the Global Precipitation Measurement (GPM) Microwave Imager (GMI) aboard the GPM core observatory are used in this study. GMI carries a series of channels from lower to higher frequencies (G-band) unlike the single-frequency CPR. Since the scattering properties depend on frequency and particle size, a combined use of CPR and GMI observations at different wavelengths has the potential to reduce uncertainties in the particle size distribution. In addition, CPR captures the backscattered echoes from hydrometers, while GMI observes extinction (absorption and scattering) signals. As shown previously (Liu, 2008), the backscattering and extinction properties change differently for various frozen particle shapes. Combining the different measurement principles of CPR and GMI may help reduce the uncertainties of particle shape. The objective of this study is to develop an algorithm to retrieve the frozen hydrometers combining CPR and GMI measurements, exploiting the frequency and instrument dependencies of the microphysical properties of ice particles.

Previous studies that have explored a combined use of a cloud radar and a microwave radiometer largely relied on simulated observations (Pfreundschuh et al., 2020) or aircraft observations (Evans et al., 2005, 2012; Pfreundschuh et al., 2022), whereas few studies analyze actual observations from multiple satellite-borne sensors used in tandem. In this study, a method is developed to retrieve the vertical profiles of IWC, number of concentration ($N_t$), mass-weighted diameter ($D_m$) and the associated uncertainties. Machine learning and optimal estimation approaches are combined into the inversion model. Section 2 details the satellite data and numerical models used in this study. Section 3 describes the methodology and flow of the retrieval algorithm. Section 4 evaluates the algorithm performance and the synergy of CPR and GMI observations. Section 5 validates the retrievals using CloudSat and GPM observations and investigates preferred assumptions of particle shape. Section 6 compares the estimates from current algorithm with existing cloud ice products. Finally, Section 7 summarizes the findings and outlines future prospects.

## 2 Data and model

### 2.1 Simultaneous observations from the GPM and CloudSat satellites

The CPR aboard the CloudSat satellite is a nadir-looking W-band radar. Table 1 outlines the specifications of the CPR. The detailed vertical structure of hydrometeors can be derived from 94GHz radar reflectivity from the CPR. The GMI aboard the GPM core satellite is a conically scanning microwave radiometer. As shown in Table 1, the GMI channels span a wide frequency range from 10 to 183GHz (Newell et al., 2015). In this study, brightness temperature ($T_b$) at frequencies of 89 GHz and higher are used since these frequencies are sensitive to the microwave scattering by frozen hydrometers. The GMI has the highest spatial resolution among space-borne passive microwave sensors equipped with frequencies above 166 GHz.



The inclined orbit of the GPM satellite has occasional orbital overlaps with polar-orbiting satellites including CloudSat, allowing for simultaneous observations at various locations from time to time.

| Freq. [GHz] | Noise (dBZ) | Vertical resolution (km) | Spatial resolution (km) |
|---|---|---|---|
| 94 | -29 | 0.5 | 1.4 |


| Freq. [GHz] | Noise (K) | Polarization | FOV (km) |
|---|---|---|---|
| 10.65 | 0.77 | V H | 20×32 |
| 18.7 | 0.63 | V H | 12×18 |
| 23.8 | 0.51 | V | 10×16 |
| 36.64 | 0.41 | V H | 10×15 |
| 89 | 0.32 | V H | 6×7 |
| 166 | 0.70 | V H | 6×6 |
| 183.31±7 | 0.56 | V | 6×5 |
| 183.31±3 | 0.47 | V | 6×5 |

**Table 1: Specifications of the GPM/GMI and Cloud Sat/CPR.**

For the evaluation of the combined GMI and CPR algorithm being developed, we utilize a match-up observation dataset from GPM/GMI, Dual-frequency Precipitation Radar (DPR) and CloudSat/CPR (Turk et al., 2021). This dataset collects data

when GPM and CloudSat fly over the same location within a time difference of 15 minutes. This dataset consists of observations from GMI, DPR and CPR along CloudSat's ground tracks as well as the collocated ECMWF atmospheric state variable data (ECMWF-AUX). For comparison, also used are an existing cloud and precipitation product (2C-ICE and 2C-RAIN) derived from CPR and Cloud-Aerosol LIdar with Orthogonal Polarization Lidar (CALIOP) aboard the Cloud-Aerosol Lidar and Infrared Pathfinder Satellite Observations (CALIPSO) satellite (Deng et al., 2015). Detailed information regarding the products and parameters used in this study is provided in Table 2. The comparison and evaluation of the

current algorithm with these datasets will be discussed in Sections 4 and 5.

| Product name | Satellite sensor | Parameter used in this study |
|---|---|---|
| ECMWF-AUX | | Pressure, temperature, specific humidity, skin temperature, surface wind 10m |



| 1C-R.GPM.GMI | GPM/GMI | Brightness temperature |
|---|---|---|
| 2A.GPM.DPR | GPM/DPR | Ku and Ka-band radar reflectivity |
| 2B-GEOPROF | CloudSat/CPR | Height, latitude, longitude, W-band radar reflectivity, CPR cloud mask |
| 2C-ICE | CloudSat/CPR and CALIPSO/CALIOP | Ice water content, effective radius |
| 2C-RAIN | CloudSat/CPR and CALIPSO/CALIOP | Liquid water content |

**Table 2: Details of the GPM, CloudSat , CALIPSO and ECMWF-AUX products.**


## 2.2 Cloud resolving model

In this study, an *a priori* database constituted of cloud and atmospheric variables is constructed with global cloud-resolving simulations from the Nonhydrostatic ICosahedral Atmospheric Model (NICAM). The development of NICAM, initially begun by (Tomita and Satoh, 2004), is currently maintained by the Atmosphere and Ocean Research Institute (AORI) at the

University of Tokyo, the Japan Agency for Marine-Earth Science and Technology (JAMSTEC), and the RIKEN Advanced Institute for Computational Science (RIKEN/AICS). NICAM has spawned numerous studies on tropical atmospheric dynamics (Miura et al., 2007; Miyakawa et al., 2014; Nakano et al., 2015). NICAM outputs of an Madden-Julian Oscillation (MJO) event offered a testbed for the assessment of cloud microphysical schemes in comparison with satellite observations (Masunaga et al., 2008). The technical details about the NICAM can be found in (Satoh et al., 2008, 2014). The version of

NICAM simulations adopted in this study was run using a single-moment microphysical scheme with a horizontal resolution of 14 km and a vertical resolution of 38 layers.

## 2.3 Forward model

The Joint Simulator for Satellite Sensors (J-sim) (Hashino et al., 2013, 2016) is used for forward simulations of satellite

observations in this study. J-sim, being developed by Japan Aerospace Exploration Agency (JAXA), contains radar and microwave radiometer modules based on the Satellite Data Simulator Unit (SDSU) (Masunaga et al., 2010), which are employed for simulating observations compatible with GPM/GMI, DPR and Cloud Sat/CPR. J-sim allows to test various microphysical assumptions such as particle size distribution (PSD) and particle shape in the forward radiative-transfer calculations (for details, see Section 3.1). Technical details of J-sim are described in Hashino et al. (2013, 2016).



# 3 Retrieval algorithm

In this section, the current algorithm methodology is described to retrieve the vertical profiles of IWC, $N_t$, $D_m$ and the associated uncertainties. The algorithm flow shown in Fig. 1 consists of two components. The first component, marked by blue dashed box, produces an initial estimation of the vertical profiles of IWC, $N_t$ (and $D_m$) using Deep Neural Network (DNN). In the second component indicated by red dashed box, Optimal Estimation Method (OEM) is adopted to optimize the frozen hydrometer profile (IWC, $N_t$ and $D_m$) using the DNN estimates as the first guess and then estimates the retrieval error. The DNN technique has the disadvantages that estimates are highly dependent on the training dataset and that uncertainty cannot be easily evaluated, but it has the advantage of obtaining reasonable estimates with a very low computational cost. The DNN technique is suitable for a quick estimation of an initial guess. On the other hand, OEM is computationally more expensive than DNN but is a well-established methodology providing statistically robust retrievals that best match observations (Rodgers, 2000) beyond the constraint of the *a priori* database used for the DNN component. OEM is suitable for the final optimization of the retrieved values and the estimation of uncertainty. The cloud microphysics assumptions commonly used by DNN and OEM are described in Section 3.1, the details of the DNN training in Section 3.2, the details of the OEM framework in Section 3.3, and an example of retrieval using this combined algorithm is shown in Section 3.4.

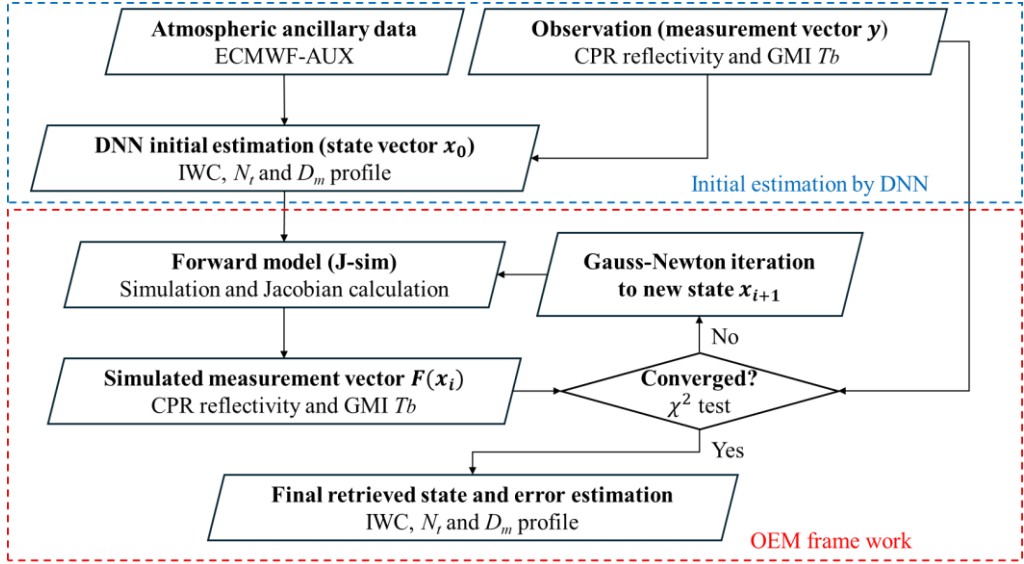

**Figure 2: Flow of the retrieval algorithm.**



### 3.1 Cloud microphysical assumption

### 3.1.1 Particle size distributions

The cloud PSD is determined in a complex manner depending on a variety of factors such as in-cloud temperature, but needs vast simplifications in practice when formulated in retrieval algorithms. In previous studies, lognormal or gamma distribution functions (Austin et al., 2009; Deng, M., G. G. Mace, Z. Wang, and H. Okamoto, 2010), which consist of temperature-dependent PSD parameters, have been mainly used to capture the basic properties of PSD. In this study, the following temperature-dependent gamma PSD function is assumed as Heymsfield and Schmitt (2013),

$$N(D) = N_0 D^\mu \exp(-\lambda D),\qquad(1)$$

$$\begin{aligned}\mu &= -14.09 - 0.248\,T\ (T < -61)\\ \mu &= -0.59 - 0.030\,T\ \ (T \geq -61)\end{aligned}\ ,\qquad(2)$$

Here $N_0$ is the intercept, $\mu$ is the dispersion, $\lambda$ is the slope parameter, and $D$ is the maximum dimension of a particle. In this algorithm, $\mu$ is prescribed as a function of temperature $(T)$ as defined by Eq. (2) of Heymsfield and Schmitt (2013), while $N_0$ and $\lambda$ are free parameters to be optimized in the algorithm. The PSD for liquid hydrometeors (cloud water and rain) is as

given by the NICAM cloud microphysical scheme (Tomita, 2008).

### 3.1.2 Particle shapes and densities

Radar and radiometric observations also depend on the shape and density of frozen hydrometeors. Frozen hydrometeors are more diverse in shape and density than liquid hydrometeors. For example, light snowflakes have a density of less than 100 kg/m$^3$ and have significantly different single scattering properties (SSP) from spherical solid ice (with the density of 916

kg/m$^3$). The Discrete Dipole Approximation method (DDA) has been widely used to calculate SSP for non-spherical particles (Draine and Flatau, 1994; Liu, 2008; Okamoto, 2002). The J-sim has an option to incorporate the SSPs of 11 different non-spherical shapes into radiative transfer calculations using pre-computed DDA databases (Liu, 2008). In addition to these non-spherical particle models, this algorithm assumes "soft sphere" with the mass - diameter (*m-D*) relationship reported in Heymsfield and Schmitt (2013). Figure 2 and Table 3 show the *m-D* relationship and the parameters of each particle model used in this study. Section 4 provides the retrieval results assuming "soft sphere" particle model, and

Section 5 discusses the optimal particle shape assumptions including non-spherical models.

| Particle shape | D$_{max}$ ($\mu$m) | Range of equal-mass sphere radius ($\mu$m) | $a_m$ (cgs units) | $b_m$ (cgs units) |
|---|---|---|---|---|
| Soft Sphere | 0-inf | 0-inf | 0.0528 | 2.1 |





| Long column | 121-4835 | 25-1000 | 0.034 | 3.0 |
|---|---|---|---|---|
| Short column | 83-3304 | 25-1000 | 0.1122 | 3.0 |
| Block column | 66-2632 | 25-1000 | 0. 2103 | 3.0 |
| Thick plate | 81-3246 | 25-1000 | 0.1064 | 3.0 |
| Thin plate | 127-5059 | 25-1000 | 0.0296 | 3.0 |
| 3-bullet rosette | 50-10000 | 19-1086 | 0.005 | 2.16 |
| 4-bullet rosette | 50-10000 | 19-984 | 0.0039 | 2.23 |
| 5-bullet rosette | 50-10000 | 21-1058 | 0.0049 | 2.23 |
| 6-bullet rosette | 50-10000 | 21-1123 | 0.0059 | 2.24 |
| Sector snowflakes | 50-10000 | 25-672 | 0.0011 | 1.54 |
| Dendrite snowflakes | 75-12454 | 33-838 | 0.0015 | 2.0 |

**Table 3: Details of the parameter for the 12 different ice-particle models.**




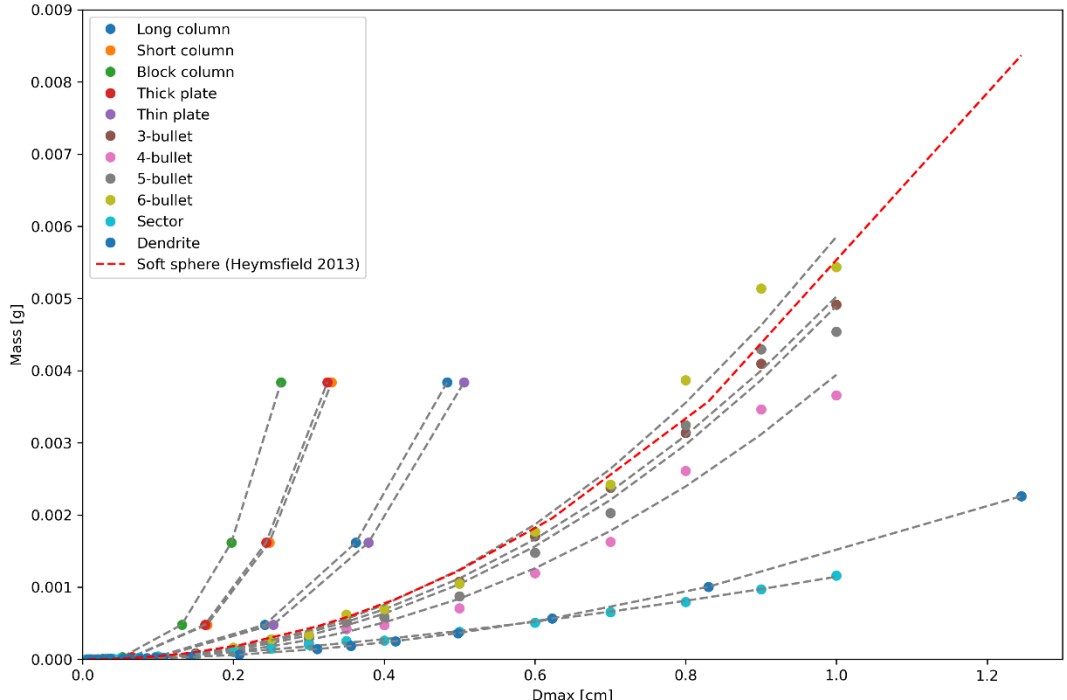

**Figure 2: Mass - diameter relationship for each particle model used in this study.**

### 3.1.3 Retrieval parameters

The retrieval parameters of frozen hydrometers are IWC, $N_t$ and $D_m$, as expressed by the following equations.

$$IWC = \int_0^\infty m(D)N(D)dD = \frac{a_m N_0}{\lambda^{\mu+b_m+1}} \Gamma(\mu + b_m + 1) \,, \tag{3}$$

$$N_t = \int_0^\infty N(D)dD = \frac{N_0}{\lambda^{\mu+1}} \Gamma(\mu + 1) \,, \tag{4}$$

$$D_m = \frac{\int_0^\infty D^4 N(D)dD}{\int_0^\infty D^3 N(D)dD} = \frac{\mu+4}{\lambda} \,,$$

$$R_e = \frac{3}{4\rho_{ice}} \frac{\int_0^\infty a_m D^{b_m} N(D)dD}{\int_0^\infty a_a D^{b_a} N(D)dD} = \frac{3a_m \lambda^{b_a-b_m}}{4\rho_{ice}a_a} \frac{\Gamma(\mu+b_m+1)}{\Gamma(\mu+b_a+1)} \,, \tag{5}$$

The definition of particle size varies among previous studies. Although $D_m$ is used in the present algorithm, effective radius

($R_e$) is also calculated by Eq. (5) for ease of comparison with existing data products. The parameters of area-diameter relationship $a_a$ and $b_b$ in Eq. (5) are set to the values reported in Heymsfield and Schmitt (2013).





### 3.2 Deep neural network for initial value estimation

The flowchart of DNN training is shown in Fig. 3. As mentioned earlier, the frozen hydrometeor datasets of the cloud resolving model (NICAM) are used as the reference data, and the observations simulated by the forward model (J-sim) from the NICAM data are input to the DNN training. The training dataset and procedure are described below in some detail.

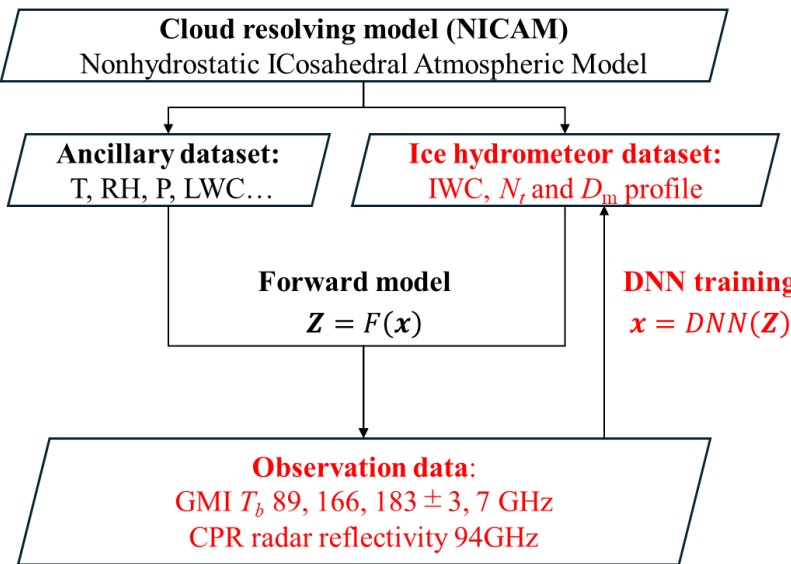

**Figure 3: Flow of DNN training using NICAM dataset.**

### 3.2.1 Training dataset

Figures 4 (a), (c), and (e) show the Contoured Frequency by Altitude Diagram (CFAD) of absolute humidity (AH), and temperature (T) from NICAM reference dataset in the tropics. For comparison, Figs. 4 (b), (d), and (f) plot the CFAD of AH and T from the ECMWF-AUX product, respectively, for three winter months (DJF) of 2015 in the tropics. The tropical oceans have little seasonal variation, so there are no significant changes over different seasons. Although not shown in Fig. 4, IWC, pressure (P), liquid water content (LWC), sea surface temperature (SST) and sea surface wind speed (SSW) obtained from NICAM are also recorded for forward calculations. Figures 4 (e) and (f) show the CFAD of radar reflectivity simulated from NICAM and actual observed CPR reflectivity. The humidity, temperature and radar reflectivity simulated from NICAM are similar to that of real atmospheric profiles, indicating that NICAM serves well as a reference (*a priori*) database for initial value estimation.




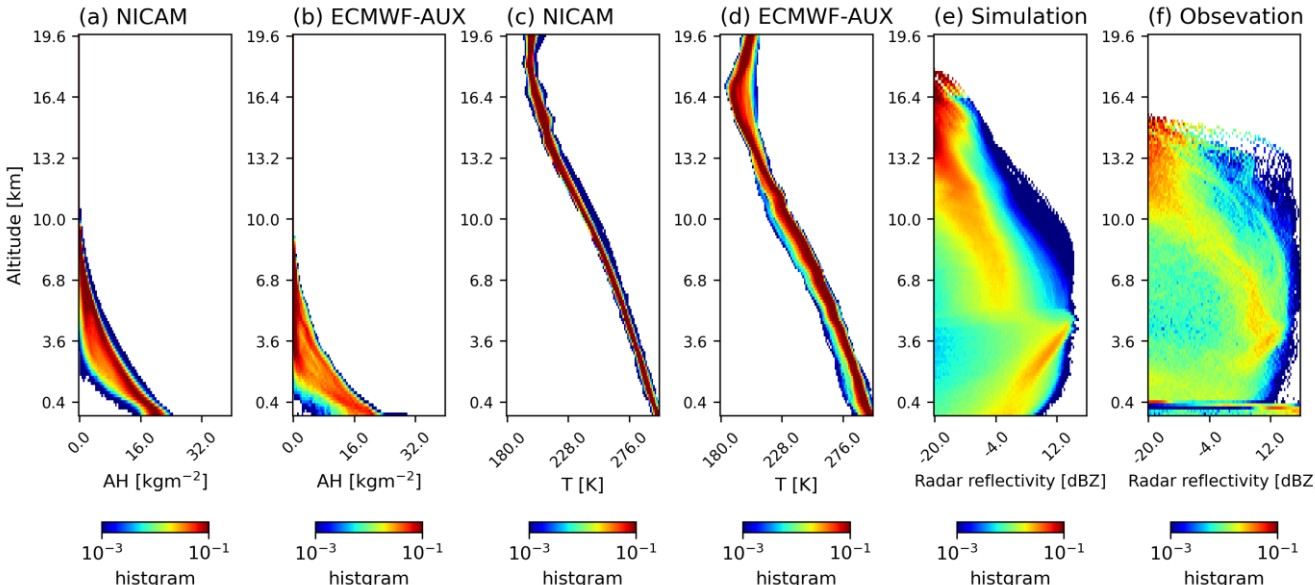

**Figure 4: CFAD of NICAM reference dataset and actual atmospheric variables. CFAD of temperature profiles for (a) NICAM and (b) ECMWF-AUX. CFAD of absolute humidity profiles for (c) NICAM and (d) ECMWF-AUX. CFAD of radar reflectivity profiles for (e) simulation and (f) CPR observation.**


### 3.2.2 DNN training

The DNN transforms the input data using the weight matrix $W$ and the activation function $\varphi(x)$. By using a nonlinear function $\varphi(x)$, DNN allows for a nonlinear transformation. In this study, the widely used ReLU function is adopted as $\varphi(x)$. The DNN inversion model $y_{DNN}$ can be written as follows for input data $x_i$.

$y_{DNN}(x_i) = W_3\varphi(W_2\,\varphi(W_1 x_i + c_1) + c_2) + c_3$ , (6)

Here, $c_n$ are constant vectors. The DNN model consists of 3 layers with 200 nodes in this study. During DNN training, $W$ is optimized using the back propagating algorithm to minimize the following loss function $J_{DNN}$.

$J_{DNN} = \sum_i(y_{DNN} - y_i)^2 = \sum_i\left(y_{DNN}(x_i) - F^{-1}(x_i)\right)^2$ , (7)

Where the reference data $y_i$ are NICAM-based frozen hydrometer profiles, and the input data $x_i$ are the simulated
observation from reference $y_i$ for GMI and CPR. Therefore, $y_i$ can be represented as the true inversion solution of forward model $F^{-1}(x_i)$. The DNN inversion model $y_{DNN}(x_i)$ would ideally approach the true inversion model $F^{-1}(x_i)$ through the minimization of $J_{DNN}$. In practice, care must be taken to avoid technical issues such as overfitting.

To stabilize the DNN training, the following preprocessing of input data is performed. GMI $T_b$ depends not only on cloud physical parameters but also on water vapor, temperature profiles, and surface emissions. To factor out these effects, $\Delta T_b$



(above 89GHz), which is the all-sky $T_b$ minus the clear-sky $T_b$, is used as the input for DNN. Clear-sky $T_b$ is obtained by repeating the forward simulations with all condensates taken out. The clear-sky $T_b$s for the real observations are calculated similarly but with the atmospheric states from ECMWF-AUX (see Section 4 for details). The CPR reflectivity profiles have a much larger number of dimensions than the GMI $T_b$ data, which are used together as DNN inputs. An Empirical Orthogonal Function (EOF) analysis is performed to retain only the first 10 principal components of radar reflectivity

profiles (EOFZe$_j$, j=1~10) so the dimension size is made comparable between CPR and GMI observations. The cumulative variance by the top 10 principal components accounts for approximately 99.9% of the total variance. The principal component $\boldsymbol{EOFZe_j}$ is obtained from the Eq. (8) and Eq. (9).

$$\frac{1}{n}\boldsymbol{ZZ^T e_j} = \lambda_j \boldsymbol{e_j} \quad (j = 1\sim10)$$
$$\boldsymbol{Z} = \left(\boldsymbol{Ze_1^{train}} - \overline{\boldsymbol{Ze_1^{train}}}, \quad \cdots, \quad \boldsymbol{Ze_n^{train}} - \overline{\boldsymbol{Ze_n^{train}}}\right) ,$$
(8)

$$\boldsymbol{EOFZe_j} = \boldsymbol{e_j^T Ze^{obs}} ,$$
(9)

Here, $\boldsymbol{e_j}$ is the eigenvector of Eq. (8), $\boldsymbol{Ze_n^{train}}$ and $\overline{\boldsymbol{Ze_n^{train}}}$ represent the radar reflectivity profile and its vertical average, respectively, for the $n$-th sample in the training data, $\boldsymbol{Ze^{obs}}$ is the observed radar reflectivity profile, and $\boldsymbol{EOFZe_j}$ is the $j$-th principal component.

It is noted that the NICAM simulations contain errors due to the limited resolution of the model or the representativeness of real cloud profiles by the model. These errors would only remotely affect the final retrieval in that the DNN-derived solution

is adjusted by the OEM as outlined next. As such, the role of the DNN in this algorithm is an efficient production of the initial values for the OEM component.

**3.3 Optimal estimation for the finale retrieval and uncertainty evaluation**

The bottom half of Fig. 1 shows the main flow of the OEM for finale retrieval and uncertainty evaluation. The OEM is a Bayesian method that finds a solution which maximize the given posteriori possibility density function $p_{post}(\boldsymbol{X}|\boldsymbol{Y})$ (Rodgers,

2000). Here, state vector $\boldsymbol{X}$ is defined by combining vertical profiles of IWC and $N_t$, and measurement vector $\boldsymbol{Y}$ is the CPR $\boldsymbol{Ze}$ and GMI $T_b$ above 89 GHz.

$$\boldsymbol{X} = \begin{pmatrix} log(IWC)_1 \\ \vdots \\ log(IWC)_n \\ log(Nt)_1 \\ \vdots \\ log(Nt)_n \end{pmatrix}, \boldsymbol{Y} = \begin{pmatrix} Ze_1 \\ \vdots \\ Ze_n \\ T_{b\,89} \\ \vdots \\ T_{b\,183\pm3} \end{pmatrix},$$
(10)

Here, $n$ is the number of cloud ice layers. IWC and $N_t$ profiles are set to be in a logarithmic form to avoid negative estimates. Assuming a Gaussian possibility density function, the cost function $J_{OEM}$ to be minimized by OEM is written as the

following equation.



$$J_{OEM} = (Y - F(X))^T S_e^{-1}(Y - F(X)) + (X - X_a)^T S_a^{-1}(X - X_a), \tag{11}$$

where $F(X)$ is the satellite observation simulated by the forward model (J-sim) from the state vector $X$, $X_a$ is *a priori* state, given by the initial value estimated by DNN, $S_a$ is the covariance error matrix for the *a priori* state. The elements of $S_a$ including off-diagonal terms are defined as $(S_a)_{kl} = \sigma_a^2 \exp\left(-\frac{d_{kl}}{L}\right)$ to take into account the level-to-level correlations

(Delanoë, J., and R. J. Hogan, 2008; Rodgers, 2000). Here, $d_{ij}$ is the distance between $k$, $l$ layer, and $\sigma_a = 0.5$, $L = 3.5$ km. $S_e$ is the covariance error matrix for measurements, which take into account not only sensor-derived measurement errors but also forward simulation-derived errors such as the uncertainty due to particle shape assumptions. The measurement error of CPR reflectivity is set to be 2.5 dBZ according to previous studies (Deng, M., G. G. Mace, Z. Wang, and H. Okamoto, 2010). A previous study (Kulie et al., 2010) reported that the uncertainty in the high-frequency $T_b$ due to particle shape assumptions

is $\sqrt{5.15}$ K at 166 GHz. In addition, since the GMI footprint is larger than the CPR footprint, errors caused by the non-uniform beam filling (NUBF) effect should be considered. The measurement error of GMI $T_b$ is set sufficiently large value of 4 K, and the off-diagonal terms of $S_e$ are assumed to be zero.

The Gauss-Newton iteration method is used as the algorithm for finding the minimum value of the cost function in Eq. (11), and the state vector of the $i$-th iteration $X_i$ is repeatedly updated until convergence according to the following equation.

$$X_{i+1} = X_i + \left(S_a^{-1} + H_i^T S_e^{-1} H_i\right)^{-1} \left[H_i^T S_e^{-1}(Y - F(X_i)) - S_a^{-1}(X_i - X_a)\right], \tag{12}$$

$$H = \begin{pmatrix} \frac{\partial Ze_1}{\partial IWC_0} & \cdots & \frac{\partial Ze_1}{\partial IWC_n} & \frac{\partial Ze_1}{\partial Nt_0} & \cdots & \frac{\partial Ze_1}{\partial Nt_n} \\ \vdots & \ddots & \vdots & \vdots & \ddots & \vdots \\ \frac{\partial Ze_n}{\partial IWC_0} & \cdots & \frac{\partial Ze_n}{\partial IWC_n} & \frac{\partial Ze_n}{\partial Nt_0} & \cdots & \frac{\partial Ze_n}{\partial Nt_n} \\ \frac{\partial Tb_{89}}{\partial IWC_0} & \cdots & \frac{\partial Tb_{89}}{\partial IWC_n} & \frac{\partial Tb_{89}}{\partial Nt_0} & \cdots & \frac{\partial Tb_{89}}{\partial Nt_n} \\ \vdots & \ddots & \vdots & \vdots & \ddots & \vdots \\ \frac{\partial Tb_{183}}{\partial IWC_0} & \cdots & \frac{\partial Tb_{183}}{\partial IWC_n} & \frac{\partial Tb_{183}}{\partial Nt_0} & \cdots & \frac{\partial Tb_{183}}{\partial Nt_n} \end{pmatrix}, \tag{13}$$

The Jacobian matrix $H$ is calculated by applying forward simulations to IWC and $N_t$ profiles being perturbed in each layer. The convergence is evaluated using the $\chi^2$ test (Rodgers, 2000) to obtain the final-retrieved state vector. OEM offers the retrieval errors defined by the trace of the following matrix $S$ defined below.

$$S = (S_a^{-1} + H^T S_e^{-1} H)^{-1}. \tag{14}$$

### 3.4 Example of the retrieval

Figure 5 shows an example retrieval for a given CPR and GMI match-up observation. The solid black lines in Figs. 5 (a) and (b) are the CPR reflectivity and GMI $\Delta T_b$ used as the input, and Figs. 5 (c) and (d) plot the DNN-based initial estimates of IWC and $N_t$ profiles and the iteration process by OEM. Figure 5 (a) also shows the CPR reflectivity and GMI $\Delta T_b$ simulated

by the forward model using the DNN and OEM estimates as the input. The DNN yields the estimates that are roughly, if not




perfectly, consistent with the satellite observations, suggesting that the DNN performs well as an initial value estimator. The OEM, refining the DNN estimate, estimates the frozen hydrometer profiles in better agreement with both the CPR reflectivity profile and GMI $T_b$. A statistical evaluation of the algorithm performance will be discussed in Section 4 and 5.

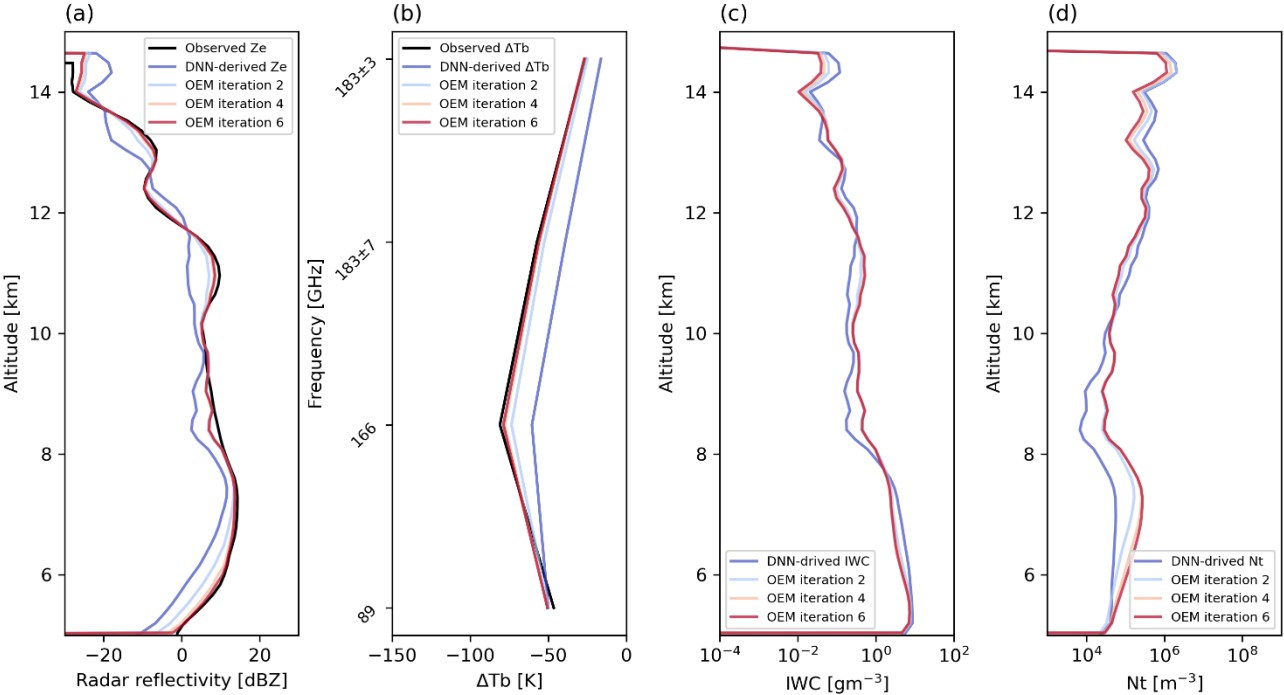


**Figure 5: Example of the initial estimation by DNN and iteration process by OEM. (a) The CPR reflectivity observations are plotted as a black line and radar reflectivity simulated from the DNN initial estimates as a dark blue line. The OEM iteration process is shown with the number of iterations, and the radar reflectivity simulated from the OEM final estimates is plotted with a dark red line. (b) Same comparison as in (a) for GMI $\varDelta T_b$. (c) The DNN initial estimates of IWC are plotted with a dark blue line and the OEM final estimates of IWC with a dark red line. (d) Same comparison as in (c) for Nt.**

## 4 Algorithm performance

### 4.1 Application to match-up observations of GPM and CloudSat

In this section, the present algorithm is applied to actual match-up observations from CloudSat and GPM satellites (Turk et al., 2021). Figures 6 (a) and (b) show a snapshot of simultaneous observations of CPR and GMI on March 18, 2016,

containing a mature tropical convective system. Observed GMI $T_b$ is plotted in solid lines and the simulated clear-sky $T_b$ from atmospheric data ECMWF-AUX is plotted in dashed lines. The GMI $T_b$ and the simulated clear-sky $T_b$ are in good agreement in the clear-sky regions (latitudes $< -11°$), showing the fidelity of the temperature and humidity sounding in use. The $\varDelta T_b$ for each channel is the difference between GMI $T_b$ and simulated clear-sky $T_b$.





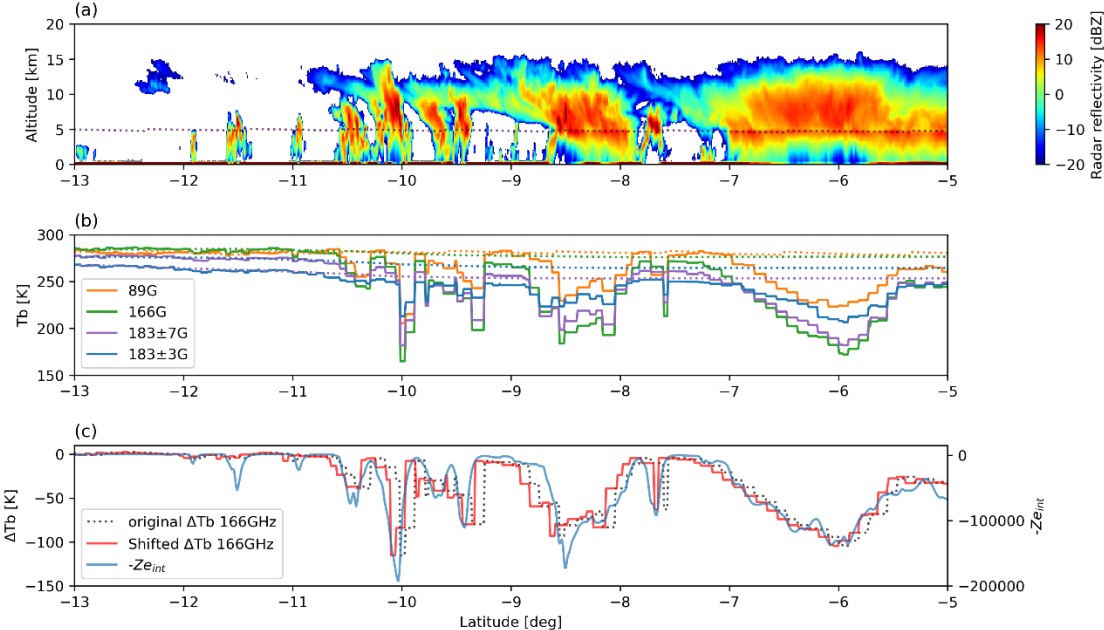

**Figure 6: The match-up observation data of CloudSat/CPR reflectivity and GPM/GMI brightness temperature for algorithm inputs. (a) The vertical distribution of CPR reflectivity and freezing level (dotted line). (b) The solid lines are GMI $T_b$ observations, and the dotted lines are the clear-sky brightness temperature simulated from ECMWF-AUX. (c) Horizontal distribution of CPR $Ze_{int}$ (red line), 166GHz $\Delta T_b$ (black line) and shifted 166GHz $\Delta T_b$ (blue line).**

Figure 6 (c) plots GMI 166GHz $\Delta T_b$ in black dotted line and vertically integrated CPR reflectivity $Ze_{int}$ defined as follows (Kulie et al., 2010) in blue line.

$$\boldsymbol{Ze_{int}} = \int_{H_{FL}}^{H_{CT}} Ze(h)dh \,, \tag{15}$$

Here, $H_{FL}$ and $H_{CT}$ are the freezing level and cloud-top height, respectively. Care needs to be taken, however, when comparing $\boldsymbol{Ze_{int}}$ with the corresponding GMI $\Delta T_b$. While the CPR is a nadir-looking radar, GMI observations have a slanted viewing angle of about 52.8 degree at Earth's surface. As a result, the layer of cloud ice aloft producing a depression of GMI $T_b$ is horizontally offset from the CPR profile matched up to the surface geolocation. To reduce the error due to this misalignment, $\Delta T_b$ is shifted so that the correlation of horizontal pattern between $\Delta T_b$ and $Ze_{int}$ becomes the highest (shown in red line). As described in Section 3.3, the errors caused by NUBF effect is already considered in the covariance matrix $S_e$. The CPR reflectivity and the shifted GMI $\Delta T_b$ are input to the algorithm to retrieve the frozen hydrometer profile. The retrieved IWC, $Nt$ and $D_m$ profiles obtained from the current algorithm are shown in Fig. 7.



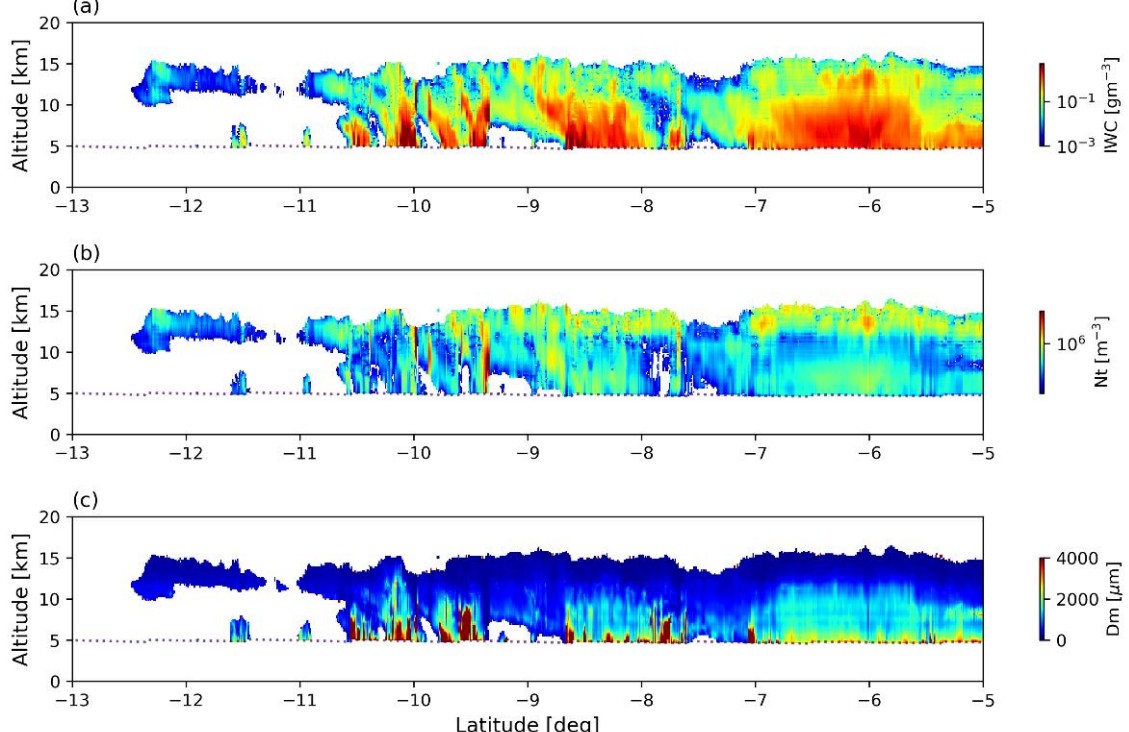

**Figure 7: The retrieved IWC (upper), Nt (middle) and Re (lower) profiles from current algorithm. The dotted lines are freezing level.**

## 4.2 Reduction of uncertainty by synergy between GMI and CPR observations

The OEM also provides the retrieval errors by Eq. (14). Figures 8 (a) and (b) show the retrieval error of IWC and $N_t$ in logarithm scale. Figures 8 (c) and (f) are example profiles of the IWC and $N_t$ retrieval errors extracted from a latitude of ~6°. To assess the performance of the GMI-CPR synergy, the retrieval errors are compared between the CPR-only (blue line) and combined-use cases (red line), respectively. The combined-use case has smaller errors than the CPR-only case in all layers, confirming a positive impact of adding GMI observations to CPR measurements.

Figures 8 (d) and (g) plot the reduction of errors when each GMI channel is added to the CPR-only observation one by one. The 89GHz $T_b$ contributes mainly to the reduction of retrieval errors in the lower layers, while 183±3GHz $T_b$ mainly reduce errors in the upper layers. The error reduction in the upper layers is exclusively owing to 183±3 GHz $T_b$ only, with the contribution of other frequencies being minimal. The 166 and 183±7 GHz $T_b$ contributes across all layers from the upper to the lower layers. Figures 8 (e) and (h) show the sensitivity of each GMI high-frequency channel to IWC and $N_t$ in each layer using Jacobian matrix in Eq. (13). The peak of error reduction shown in Figs. 8(d) and (g) is consistent with the peak of sensitivity shown in Figs. 8(e) and (h) for each channel.



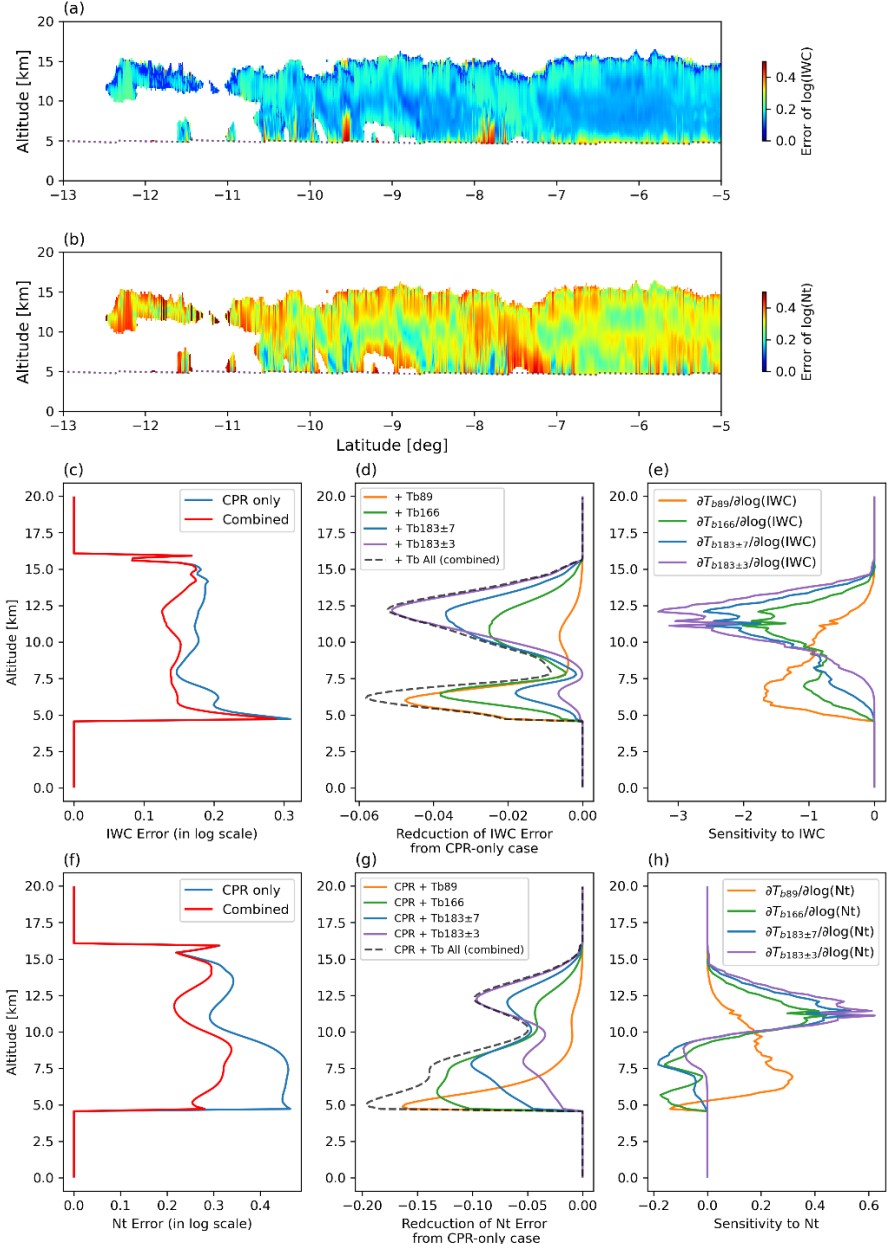


**Figure 8: Retrieval error analysis for investigation of synergy between CPR and GMI observations. Retrieval error profiles of (a) IWC and (b) $N_t$ in logarithmic scale. (c) An example of IWC error profiles calculated for CPR-only (blue line) and combined-use case (red line). (d) Error reductions of IWC from the CPR-only case by adding each GMI high-frequency channel to the CPR observation. (e) Sensitivity (Jacobian) of each GMI high-frequency channel to IWC in each layer. (f) Same comparison as in (c) for**
**$N_t$. (g) Same comparison as in (d) for Nt. (h) Same comparison as in (e) for Nt.**





# 5 Consistency in measurement space

## 5.1 Reproducibility of CPR and GMI observations

In-situ data to validate cloud physical parameters are limited in availability. The algorithm performance is therefore tested
using measurables ($Ze$ and $T_b$) instead of retrieved variables (IWC, $N_t$ and $D_m$). To this end, measurables are reproduced with
forward simulations using the retrieved frozen hydrometer parameters as the input for comparison with actual observations.
This comparison is performed using 10 match-up observations of CPR and GMI, including the case shown in Figs. 6 and 7.
The 2C-RAIN product is used for cloud liquid water and rain water beneath the cloud-ice layer. As far as the layer of liquid
cloud and rain is optically thick for microwave radiation as typical of heavily raining clouds, high-frequency $T_b$ becomes far
less sensitive to LWP than to IWP (Masunaga, 2022) and the uncertainty resulting from the liquid component is negligible.
Figures 9 (a) and (b) show an example of simulated CPR radar reflectivity in the solid-phase layer and GMI $T_b$ from the
current algorithm estimates. Compared to the actual observation shown in Fig. 6, the spatial structure of radar reflectivity
and the horizontal distribution of $T_b$ are reproduced well. Figures 9 (c) and (d) are scatter plots of the simulated and actual
observations for the 10 match-up cases. The simulated radar reflectivity and $T_b$ at high-frequency channels are both overall
unbiased against the actual observations. This result assures self-consistency of the current algorithm.







**Figure 9: Reproducibility of the CPR reflectivity and GMI $T_b$ for the current algorithm. (a) Example of the simulated CPR reflectivity and (b) simulated GMI $T_b$ (solid lines) from the frozen hydrometeors estimated by the current algorithm. Dotted lines are actual GMI $T_b$ observations for comparison. (c) Statistical comparison between actual reflectivity and simulated reflectivity for 10 match-up cases. (d) Results of the same comparisons as in (c) for each GMI high-frequency channel.**




## 5.2 Reproducibility of DPR observations

DPR carried by the GPM core observatory yields simultaneous observations with GMI radiometry, providing additional data to assess the algorithm performance. GPM/DPR, a suite of Ku- and Ka-band radars, are sensitive to large frozen hydrometeors such as snow and graupel inside of deep convective clouds, having information independent of CPR and GMI observations. This study uses DPR reflectivity above freezing level to test the cloud ice estimates from the present algorithm. Similarly to Fig. 9, the Ku- and Ka-band radar reflectivity are simulated from the current algorithm estimates of frozen hydrometers (Figs. 10, b and d) for comparison with the actual DPR observations ((a) and (c)). The current estimates of cloud ice reproduce the overall distribution of observed Ku and Ka radar reflectivity. As shown in Figs. 13 (e) and (f), the simulated DPR reflectivity exhibits no systematic bias against the actual DPR observation for the 10 match-up cases despite the significant spread.





**Figure 10: Reproducibility of DPR Ku and Ka-band radar reflectivity. Snapshot of the (a) Ku-DPR observation and (b) simulated Ku-band reflectivity from the current algorithm estimates. Snapshot of the (c) Ka-DPR observation and (d) simulated Ka-band reflectivity from the current algorithm estimates. (e) Statistical comparison between actual Ku-DPR observations and simulated Ku-band reflectivity using 10 match-up cases. (f) Same comparison as in (e) for Ka-band reflectivity.**




### 5.3 Particle shape assumptions

This section discusses the assumptions of the particle model optimal for this synergistic algorithm. CPR reflectivity mainly captures the backscattering properties of particles, while GMI $T_b$s mainly observe the scattering and absorption properties. A combined use of these two independent information has the potential to constrain uncertainties in the assumptions of the particle model. To test this, the reproducibility of CPR and GMI observations are evaluated with different non-spherical particle models listed in Table. 3. Only the particle model that consistently represents all the backscattering, absorption, and

scattering properties would allow the algorithm to find the solution (frozen hydrometer profile) that accords with both CPR and GMI observations. Figures 12 (a)-(f) compare the simulated reflectivity and $T_b$ with the actual CPR and GMI observations for the six representative particle models (long column, thin plate, 4-bullet rosette, sector snowflake, dendrite snowflake and soft sphere). The CPR reflectivity is well reproduced regardless of the particle model assumptions by optimizing IWC and $N_t$ (that is, the PSD parameters $\lambda$ and $N_0$) in our algorithm. On the other hand, the simulated $T_b$ is

clearly lower than the observed $T_b$ for cold $T_b$s, except for the dendrite snowflake and soft sphere cases. These results indicate two points: 1) the soft sphere and dendrite snowflake are the optimal particle models among the six models tested here, and 2) CPR observations alone are not sufficient to simultaneously constrain the uncertainties in the PSD and particle models.







**Figure 11 Comparison of the reproducibility of CPR and GMI observations for various particle models. Scatter plots between actual observations and simulated observations assuming (a) long column, (b) thin plate, (c) 4-bullet rosette, (d) sector snowflake, (e) dendrite snowflake and (f) soft sphere. (g) Dependency of $T_b$ bias (simulation – observation) of $Ze_{int}$ (IWP) for each particle model.**




Figure 11 (g) plots the difference between the simulated and actual $T_b$ as a function of IWP for each particle model assumption. Since the IWP estimate varies with the particle model assumptions, the horizontal axis is substituted by $Ze_{int}$ in Eq. (15). Simulated $T_b$ is much lower than the observation for most non-spherical particle models at large $Ze_{int}$ , whereas for the dendrite snowflake and soft sphere, $T_b$ bias is relatively small for the whole range of $Ze_{int}$. A previous study (Fig. 11 in

Kulie et al., 2010) shows a similar figure to Fig. 12 (g) and  also reported that most non-spherical particle models except the dendrite snowflake exhibit excessive scattering (negative biases to actual observation) for large IWPs. Kulie et al. (2010) first converted CPR reflectivity to IWC assuming the Liu's non-spherical model (Liu, 2008), and then simulated $T_b$ from this IWC assuming the same particle model to compare with the actual SSMIS 157GHz $T_b$. Their study assumed a fixed PSD when simulating $T_b$, so the failure to reproduce GMI $T_b$s may be caused by an inappropriate PSD assumption rather than the

particle model. However, even our algorithm, which optimizes PSD parameters by OEM, cannot find the solution that is simultaneously consistent with the CPR and GMI observations for non-spherical particle model except for dendrite snowflake.

In Fig. 12 (g), the soft sphere assumption best reproduces the satellite observations, while many previous studies have shown

that non-spherical particles are more appropriate assumption than soft sphere particle models (Ekelund and Eriksson, 2020; Galligani et al., 2015; Kulie et al., 2010). The reason for the disagreement with previous studies may be mainly due to the difference in the assumed *m-D* relationship. Since the density affects the effective dielectric constant of the particles according to the Maxwell-Garnett formula, the optical properties of the soft sphere depend on the assumed *m-D* relationship. Liu et al. (2004) shows that the optical properties of any non-spherical particle model can be approximated by a soft sphere

with the density appropriately adjusted. That is, the result of Fig. 12 (b) shows that the soft sphere with the *m-D* relation (Heymsfield and Schmitt, 2013) used in this algorithm may be able to approximate a certain optimal particle model that is most consistent with real satellite observations in this case. It should be noted, however, that this result does not tell us whether the particles are actually spherical or not, even though the soft sphere is used as a proxy of real frozen particles. Since real clouds are a mixture of various particle shapes and the dominant particle shape varies with altitude, we should

investigate using mixed non-spherical particle models, but this is a complex task and should be discussed in the future.

## 6 Comparison with other cloud and precipitation products

The current retrieval is compared with the CloudSat/CALIPSO standard radar/lidar product (2C-ICE). Figures 12 (a) and (b) show the IWC and $R_e$ estimates of 2C-ICE, and the observable areas for the CALIPSO lidar (lidar cloud mask above 25%) is shaded in grey. Figures 12 (c) and (d) compare the 2C-ICE estimates with the current algorithm assuming soft sphere for the

10 match-up cases. Here, $R_e$ is calculated using Eq. (5). The IWC estimates of current algorithm agree very well with 2C-ICE, but there is a positive bias in $R_e$. In particular, the $R_e$ bias tends to increase with $R_e$ toward deep inside the cloud layer.




As shown in Figs. 12 (a) and (b), in the 2C-ICE product, the combined radar and lidar observations are limited to near the cloud tops, so the retrieval in deeper cloud layers is almost based on the CPR observations only. On the other hand, this study also uses GMI, allowing synergetic observations even deep inside of clouds (as shown in Figs. 8 (e) and (h)). The current algorithm actually captures large snow and graupel particles inside convective clouds, to which the DPR is sensitive (as shown in Fig. 10). In addition, lidar is sensitive to small particles, whereas microwave instruments are sensitive only to relatively large hydrometers. One possible reason for the $R_e$ bias is the difference in the sensor-specific sensitivity. Other factors could be differences in the cloud microphysical assumptions such as PSD and particle shape.


**Figure 12: (a) Example of IWC and $R_e$ profiles of 2C-ICE product in same case as Fig. 6 and 7. (b) Comparison of IWC and $R_e$ between 2C-ICE and the current estimates assuming soft sphere for 10 match-up cases.**



## 7 Summary


This study develops an algorithm to retrieve the vertical profiles of IWC, $N_t$ and $D_m$ in deep convective systems using simultaneous CPR and GMI observations. A new algorithm that combines the DNN and OEM for the inversion problem solver is proposed. The role of DNN in this algorithm is to estimate near-optimal initial values at low computational cost. The DNN is trained using *a prior* database constructed from the cloud resolving model (NICAM) (Fig. 4). The OEM uses

the DNN estimate as an initial state to further optimize the frozen hydrometer profile to be consistent with CPR and GMI observations (Fig. 5). The retrieval error is calculated as a byproduct of the OEM at the in same time. The retrieval performance of the current algorithm is evaluated using match-up observations of CPR and GMI (Figs. 6 and 7). The combined use of CPR and GMI reduces the retrieval error compared to the case using CPR only, indicating a positive impact of the synergy between CPR and GMI observations (Figs. 8 (c) and (f)). These reductions of retrieval error are significant at

multiple altitudes where the GMI high-frequency $T_b$ is most sensitive to ice particles (Figs. 8 (d), (e), (g), and (h)).

To evaluate the validity of the current algorithm estimates, the reproducibility of microwave $T_b$ and radar reflectivity is tested through forward simulations. The CPR and GMI observations are overall reproduced to a reasonable extent (Fig. 9). Furthermore, the current estimates statistically reproduce the DPR observations (Ku- and Ka-bands), which have independent information and are sensitive to large snow and graupel particles inside convective clouds (Fig. 10). In addition,

it was found that the evaluations of the simultaneous reproducibility of CPR reflectivity and GMI $T_b$s can constrain the choice of non-spherical particle model. For dendrite snowflake and Heymsfield's soft sphere, $T_b$ bias is relatively small regardless of IWP, whereas the simulated $T_b$ is much lower than observed $T_b$ at large IWP for other particle models tested (Fig. 11).

Finally, the current estimates are compared with the existing radar-lidar cloud ice product (2C-ICE) (Fig. 12). The results are

statistically in agreement for IWC, but Re tends to be overestimated by the current algorithm compared to 2C-ICE. The biases may be caused by differences in cloud microphysics assumptions (such as particle models) and the sensitivity of the sensors used in the algorithm.

The framework of the algorithm developed in this study can be applied to the combined use of various cloud/precipitation radars and millimeter/submillimeter radiometers by adjusting the sensor configuration of the forward model. In the future,

we plan to extend the algorithm to Doppler CPR carried by the EarthCARE satellite and millimeter/submillimeter-wave radiometers such as GOSAT-GW/AMSR3 and MetOp-SG/ICI, which are to be launched within the next few or several years.

**Code availability**

Available upon request.



## Data availability

The match-up observation dataset from GPM/GMI and CloudSat/CPR are include in (Turk et al., 2021).

The forward model used in this study (Joint Simulator for Satellite Sensors) is available from https://www.eorc.jaxa.jp/theme/Joint-Simulator/userform/js_userform.html.

The NICAM data used in this work will be made available upon request by the authors.

Tensorflow module in python are used for machine learning (Deep Neural Network).

## Author contribution

All coding, analysis, and writing of the first draft of this study were performed by KO. The study design, discussions on results, and manuscript improvement were greatly assisted by HM.

## Competing interests

The authors declare that they have no conflict of interest.

## Acknowledgements

Prof. Guosheng Liu and Assoc. prof. T. Hashino are gratefully acknowledged for making the non-spherical scattering database publicly available and helping us use it in the forward model (Joint Simulator for Satellite Sensors). The authors thank Prof. Kentaro Suzuki for providing the data of cloud resolving model (NICAM) for our research.

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
