# Peer review of "Synergy of millimeter-wave radar and radiometer measurements for retrieving frozen hydrometeors in deep convective systems"

_EGUsphere, 2025_

## Author Response (AR1)

Dear Referee,

We would like to express our sincere gratitude to the reviewers for their careful reading of our manuscript and for providing valuable and insightful comments. We have carefully considered and responded to each point below and revised the manuscript accordingly based on these responses. A list of changes made in the manuscript (track-changes file) is described in red text. We hope that our explanations and revisions address all the concerns raised and meet the reviewers' expectations.

Referee 1

Referee major comment 1:
The "a priori" covariance S_a. It is unclear why the authors use a formulation used in previous studies when the DNN model provides the "a priori" estimates. Given that the DNN retrievals were developed using simulations, the authors could evaluate retrieval errors using an independent simulated dataset (or setting aside a fraction of the existing simulated dataset for evaluation) and calculate the associated S_a. This should be discussed in the manuscript.

Author response:
As you rightly pointed out, it is mathematically appropriate to use the estimation error of the DNN to construct the a priori error covariance matrix S_a.
Directly estimating the retrieval error (i.e., the diagonal elements of S_a), however, is generally difficult for a typical DNN, and more advanced methods such as Quantile Regression Neural Networks (QRNN) (Amell et al., 2022) are required. Estimating the off-diagonal elements of S_a, which represent the error correlations between different layers, is an even more challenging. We consider the estimation of S_a using DNNs to be an intriguing topic that could be pursued in future work.
It might be also possible to construct S_a from the a priori dataset (i.e., the DNN training data). We use the cloud-resolving model NICAM as the a priori dataset in this study. That being said, since NICAM (or any other numerical model) is a limited representation of cloud statistics in the real atmosphere, it is not clear whether we can fully trust the error correlations derived from NICAM. For these reasons, we prefer a simple approach of treating S_a (specifically, $\sigma$_a and L) as a tuning parameter to be determined experimentally by varying $\sigma$_a in the range of 0.25 to 1 and L in the range of 1 to 10. Although the results don't change significantly, we select the values that yield the best retrieval performance (i.e., best agreement with the observations). We incorporated this discussion into the manuscript (lines 264-268, 270-271).

Amell, A., Eriksson, P., and Pfreundschuh, S.: Ice water path retrievals from Meteosat-9 using quantile regression neural networks, Atmos. Meas. Tech., 15, 5701–5717, https://doi.org/10.5194/amt-15-5701-2022, 2022.

Referee major comment 2:

The interpretation of results via Eq. (14). Specifically, the authors state that matrix S in Eq. (14) provides the error of the estimated variables. While this may be considered true at some general (and approximate) levels, S is more rigorously the posterior error covariance. If the "a priori" error covariance S_a is correctly estimated and the forward modelling errors are correctly specified, S is indeed the true error covariance. However, given that both S_a and the modeling errors may not be accurately estimated, covariance S given by Eq. (14) could be significantly different from the actual error covariance. Moreover, theoretically, the inclusion of observations always results in a smaller S, but practically the reduction in S depends on how accurate the forward models are. Therefore, the authors should clarify that the results shown in Fig. 8 are not errors in the true sense (estimate-true) because the true values are unknown. Instead, these results are theoretical estimates derived using Eq. (14), and this limitation should be discussed.

Author response:
As you correctly pointed out, this study does not provide rigorous estimates of the a priori error covariance matrix $S\_a$ and of the forward modeling error. The retrieval error covariance $S$, derived from Eq. (14), may therefore not accurately represent the true retrieval error. Care should be taken when interpreting the results shown in Fig. 8, which are based on these theoretically estimated errors.

Since the true retrieval error is unknown, Fig. 8 is not intended for a quantitative assessment of the reduction in error by including brightness temperature observations. Figs. 8 (d), (e), (g), and (h) nonetheless meet physical expectations in a qualitative sense in that the $183 \pm 3$ GHz brightness temperature reduces estimated errors in upper-level cloud ice, while the 89 GHz channel reduces errors in lower-level cloud ice. We revised the manuscript (lines 287-288, 349-353) to incorporate these discussions.

Referee major comment 3:
The performance of the soft-sphere electromagnetic calculations is somewhat surprising. While soft-sphere calculations have been shown to work in some cases, it has also been shown that it is generally difficult (or impossible) to find assumptions about the density of hydrometeors that work for a wide range of frequencies (Kuo et al., 2016; Olson et al., 2016). The backscattering properties of snow particles at W-band differ significantly from those of soft spheroids except for an equivalent density of 0.3 g/cm^3. Therefore, the fact that soft spheroids result in the best agreement should not be construed as a general indication that the soft-spheroid approach works in all cases. This is especially true given that the largest discrepancies occur at the low end of the brightness temperatures and that

the radar model does not account for multiple scattering. This limitation needs to be acknowledged and discussed.

Author response:
We agree that "the fact that soft spheroids result in the best agreement should not be construed as a general indication that the soft-spheroid approach works in all cases.". Figure 11(g) shows that the soft-sphere assumption leads to the best reproducibility of brightness temperatures and radar reflectivity for clouds with large IWP, such as deep convective clouds. For clouds with moderate or smaller IWP, on the other hand, there is little difference in brightness temperature reproducibility among the tested particle models. In other words, it remains unclear whether the soft-sphere assumption is optimal for more common thin ice clouds. To avoid misunderstanding, we revised the manuscript (lines 21-23, 413, 426-430) to clearly state that the soft-sphere assumption is optimal for tropical deep convective clouds with large IWP but is not otherwise.

As you pointed out, the results may be also influenced by several factors not yet considered, such as multiple scattering effects on radar reflectivity and the presence of supercooled liquid water. When these effects are fully considered, it remains uncertain whether the soft-sphere assumption would still be the most appropriate even for clouds with large IWP. We would like to regard these issues as important topics for future study.

We also acknowledge the fact that "it is generally difficult (or impossible) to find assumptions about the density of hydrometeors that work for a wide range of frequencies (Kuo et al., 2016; Olson et al., 2016)." As demonstrated by Liu et al. (2004), the scattering properties of various nonspherical particles can be approximated by varying the density of spherical particles. However, the best-fit density depends on frequency, making it difficult to approximate the scattering properties of nonspherical particles across a wide frequency range with a single-density sphere. The following two hypotheses may explain the reasonable performance of soft spheres for large IWPs:

(1) As previously discussed, the soft-sphere assumption may have been appropriate for deep tropical convective clouds with very large IWP. This may be because an appreciable amount of graupel is formed by riming in deep convection. The scattering properties of graupel are likely better approximated by soft spheres than snowflakes and ice crystals. Olson et al. (2016) focused on stratiform precipitation, where scattering was likely dominated by aggregated snow particles, making the soft-sphere assumption less appropriate. The differences in cloud microphysics between convective and stratiform clouds may explain the contrasting results between the present and previous studies. We plan to apply our method to a wider range of cases in future work to further investigate this topic.

(2) The validity of soft spheres may vary largely with the particle density model (*m-D* relation) in use. The present work adopts the *m–D* relationship from Heymsfield and Schmitt (2013), which is different from the soft sphere model used in previous studies. Our study (Fig.11 (g)) suggests that this soft sphere model performed better than nonspherical particles from Liu (2008). The current finding does not imply that all soft sphere models, if any, are superior to non-spherical particle models used in previous studies. It also remains possible that other nonspherical particles, such as those used in Kuo et al. (2016) and Olson et al. (2016), would yield even better results. We plan to incorporate a range of nonspherical particle models, such as aggregated snow particles, in future studies.

We added the suggested references and reflected these important discussions in the revised manuscript (lines 440-466).

Referee minor comment 1:
Eqs. (1) and (2). Delanoe et al. (2014) use a different formulation in which the shape (mu) dependence of the integrated properties is not a important as that of the generalized intercept that can be parameterized as a function of temperature. The normalized PSD approach is likely to explain better variability in the PSD with a reduced number of parameters.

Author response:
Thank you for the valuable information. We would like to try to use the normalized PSD in future work.

Referee minor comment 2:
How is H in Eq. (13) calculated (i.e. finite-difference or automatic differentiation)?

Author response:
The Jacobian matrix H is calculated using the finite difference method by perturbing the IWC and Nt for each layer and performing iterative forward calculations. A supplementary explanation will be added to the main text (line 283).

Referee 2

Referee major comment 1:
At these higher (89 GHz and higher) frequencies, the attenuation due to water vapor is significant. In the tropical regions, the attenuation due to water vapor is significant (up to 2-way path attenuation exceeding 8 dB; see Josset et al. 10.1109/TGRS.2012.2228659). And for radiative transfer at 89, 166 and the various 183 GHz water vapor bands, the amount and vertical extent of the water vapor can reduce the overall scattering albedo and impact simulation of TB at these channels. My question is: How "accurate" is the specification of the ancillary data used (ECMWF-AUX)? In your figure 2, these data appear to be used as a one-time "fixed" input, indicating that the water vapor profile stays fixed while you vary the ice particles in the forward OEM simulations. Would you expect the water vapor profile to be the "same" across different types of ice particle shapes (dendrite, long column, etc.)? While I am no expert in this topic, in nature water vapor and ice particle processes are likely correlated to some extent.

Author response:
We use the water vapor profile from ECMWF-AUX as a "fixed" input and optimize only the ice cloud profile. As shown in Figure 6 (b), the GMI Tb under clear-sky conditions is well reproduced by the simulated Tb using ECMWF-AUX, indicating a certain degree of confidence in the accuracy of ECMWF-AUX.

Ideally, the water vapor profile should also be included in the state vector $X$ of Eq. (10) and optimized by OEM framework. From a technical perspective, however, optimizing both the ice particle and water vapor profiles is computationally demanding, as the amount of information provided by satellite observations (i.e., the dimension of the observation vector $Y$ in Eq. (10)) is too small relative to the number of unknown parameters to be retrieved (i.e., the dimension of the state vector $X$). This would cause convergence issues in the retrieval. For thick ice clouds such as deep convective clouds, scattering signals from ice particles are expected to dominate brightness temperature despite the considerable absorption and emission signals from water vapor.

We also conducted sensitivity tests assuming 100% or supersaturated humidity within the cloud, which showed little impact on the retrieval results. This practically justifies the simplified approach of optimizing only the cloud ice profile with a given water vapor profile in this study. We have added an explanation in the main text (lines 251-259, 311-314). We plan to revisit it in future studies using additional satellite observations, such as Doppler radar and submillimeter-wave measurements.

We are aware that ice particle habits correlate with temperature and supersaturation. Assuming or mixing different particle shapes depending on the water vapor and temperature profiles will also be an important direction for future research.

Referee major comment 2:
The GMI has dual-polarized (V and H) capabilities at 89 and 166 GHz. Previous studies have indicated polarization difference especially at 166 GHz (Gong et al. 2017, https://doi.org/10.5194/acp-17-2741-2017). In your forward simulation, were polarized TB calculations performed? The extent of V-H polarization difference may provide additional independent information to identify and/or constrain the type of ice particles appropriate for certain deep convective clouds.

Author response:
In the forward model used in this study, we assume randomly oriented particle models and therefore do not account for the differences between the V- and H-polarized brightness temperatures (Tb) caused by ice particle orientation. The use of Tb polarization differences is a very interesting topic, and we would like to address it as a subject for future research. We added a note in the manuscript (lines 167-169) to clarify that the effects of ice particle orientation were not considered in this study.

Referee minor comment:
Just FYI- The CloudSat-GPM (and CloudSat-TRMM) dataset has recently been updated to cover all current Release-5 CloudSat data. While the data products themselves remained unchanged, the data cover up thru mid-2020. Details are available at NASA's Precipitation Processing System (https://arthurhou.pps.eosdis.nasa.gov) and details at: https://arthurhou.pps.eosdis.nasa.gov/Documents/CSAT_TRMM_GPM_COIN_ATBD_V 05.pdf.

Author response:
We would like to use the updated dataset in future work.

Other minor revisions

Lines 146, 390, 408, 431: Figure numbers were corrected.

Figure 11: Figure title "(g)" was added.

Line 435: "GMI" was corrected to "SSMIS".

Line 468: Section title was changed.

Lines 562-567, 585-588, 620-623: Reference was corrected and added.

Sincerely,
Keiichi Ohara